# Color Names in Vision-Language Models

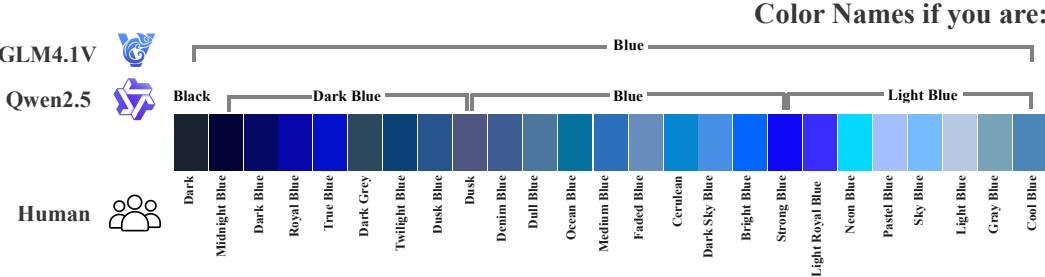

Figure 1: Color naming precision varies dramatically across systems. When shown blue color strips, GLM4.1V demonstrates limited color vocabulary, labeling diverse hues with the single term 'blue', while Qwen2.5 shows moderate discrimination using modifiers like 'light blue' and 'dark blue'. In contrast, humans exhibit rich color vocabularies with distinct names for each perceptual variation. Human terms from Lindner et al. (2012).

## Abstract

Color serves as a fundamental dimension of human visual perception and a primary means of communicating about objects and scenes. As vision-language models (VLMs) become increasingly prevalent, understanding whether they name colors like humans is crucial for effective human-AI interaction. We present the first systematic evaluation of color naming capabilities across VLMs, replicating classic color naming methodologies using 957 color samples across five representative models. Our results show that while VLMs achieve high accuracy on prototypical colors from classical studies, performance drops significantly on expanded, non-prototypical color sets. We identify 21 common color terms that consistently emerge across all models, revealing two distinct approaches: constrained models using predominantly basic terms versus expansive models employing systematic lightness modifiers. Cross-linguistic analysis across nine languages demonstrates severe training imbalances favoring English and Chinese, with hue serving as the primary driver of color naming decisions. Finally, ablation studies reveal that language model architecture significantly influences color naming independent of visual processing capabilities.

## 1 Introduction

Vision-language models (VLMs) have rapidly evolved from tasks like image captioning and visual question answering (VQA) to becoming core tools for evaluating multimodal AI systems (Huang et al., 2025). Their integration into commercial large language models (LLMs) such as ChatGPT and Claude has democratized access to multimodal capabilities, enabling millions to use them for everyday visual queries.

To support this widespread use, the VLM community has developed benchmarks for general tasks (e.g., VQA, OCR, image captioning (Li et al., 2025).) and more specialized ones targeting failure modes like hallucination (Guan et al., 2024; Li et al., 2023) or spatial awareness (Zhang et al., 2025). Yet, color remains underexplored—existing evaluations

focus only on broad color categories, lacking systematic analysis of color vocabulary and naming consistency.

This oversight is critical, as users often expect accurate color descriptions in image-based interactions (Chatterji et al., 2025). Color is a core aspect of human visual perception and communication (Berlin & Kay, 1991; Witzel & Gegenfurtner, 2018), making its evaluation essential for effective human-AI interaction.

In this work, we present the first systematic study of color naming in VLMs. Rather than assessing precise color metrics, we adopt a categorical naming approach rooted in color perception research (Witzel & Gegenfurtner, 2018), reflecting how humans naturally describe color (e.g., light red, pink, yellow). This aligns with the linguistic interface VLMs must navigate when describing visual scenes.

Following cross-linguistic color studies (Berlin & Kay, 1991), we evaluate VLMs using uniform color samples to isolate intrinsic color naming capabilities. While this abstracts away real-world complexity, it avoids confounds like object identity and lighting, establishing a clear baseline for future, context-rich evaluations.

Our contributions are the following:

- We perform the first color naming anaysis in VLMs replicating classic color name methodology: while VLMs achieve high accuracy (94-98%) on prototypical colors from classical studies, their performance drops significantly when evaluated on expanded, non-prototypical color sets.
- Interestingly, 21 common color terms emerge consistently across all evaluated VLMs and reveal two distinct approaches: constrained models using predominantly basic terms versus expansive models employing systematic lightness modifiers for fine-grained color discrimination.
- Mutual information analysis reveals that hue serves as the primary driver of color naming decisions across all models. However, the percentage of hue, saturation and value explains color names in different proportions for different models, suggesting different encoding strategies.

## 2 Motivation: Basic color categories

Classic color naming studies (Berlin & Kay, 1991; Sturges & Whitfield, 1995) established that humans universally organize color space using eleven basic categories (black, white, red, green, yellow, blue, brown, purple, pink, orange, gray). To test whether VLMs exhibit similar categorization, we replicated Berlin and Kay's experiment using their 330 Munsell chips on five representative VLMs (full experimental details, including prompt, are in supplementary material).

Table 1 shows that VLMs achieve high accuracy (94- 98%) when compared to human focal color data (Sturges-Whitfield), suggesting strong alignment with universal color categories. However, this high performance reflects evaluation only on a subset (111 chips) of highly saturated, prototypical colors, which are precisely the conditions where human agreement is strongest.

When evaluated against the complete 330-chip Munsell set using computational color models (NICE(Parraga & Akbarinia, 2016) and Benavente(Benavente et al., 2008)), accuracy drops to 70-83%, revealing systematic deviations from optimal color naming on non-prototypical colors. This limitation becomes critical when we consider that real-world color naming must handle the full perceptual spectrum, including desaturated, intermediate, and atypical colors where systematic naming patterns are less established. Berlin and Kay's focal approach, while foundational, cannot reveal whether VLMs develop coherent color vocabularies across the complete range of perceivable colors.

Nevertheless, the previous analysis on Berlin and Kay's approach already gives a hint at how problematic color naming becomes when considering non-prototypical colors. Thus, to perform a more in-depth analysis of color naming in current VLMs, in the remainder of

Table 1: Model Accuracy Across Human Color Boundary Datasets. Sturges-Whitfield is computed on only 111 chips. NICE and Benavente use the full 330-chip set.

| Model | Sturges-Whitfield | NICE | Benavente |
|---|---|---|---|
| InternVL2.5 8B | 0.942 | 0.738 | 0.700 |
| Qwen2.5 7B | 0.827 | 0.647 | 0.647 |
| Molmo 7B | 0.981 | 0.816 | 0.831 |
| JanusPro 7B | 0.981 | 0.809 | 0.812 |
| GLM4.1V 9B | 1.00 | 0.775 | 0.762 |
| MiniCPM V4.5 | 0.875 | 0.637 | 0.656 |

the paper, we follow an experimental setup that considers a $3\times$ larger set of color samples. This expanded dataset enables a more comprehensive investigation of naming consistency, coverage, and sensitivity to non-prototypical colors.

## 3 Experimental Setup

### 3.1 Dataset

For systematic evaluation of VLM color naming, we utilize the 957 color samples employed by Lindner et al. (2012) in their large-scale multi-lingual color analysis. This dataset provides comprehensive coverage of perceivable color space, spanning both common and uncommon color variations across the full spectrum of human color perception. The color samples were originally derived from the XKCD color survey Munroe (2010) and have been validated through cross-linguistic color naming studies Lindner et al. (2012).

### 3.2 Models

We evaluate color naming behavior across six representative vision-language models from different architectural families: GLM4.1V 9B (GLM-V-Team et al., 2025), MiniCPM-V4.5 8B (Yu et al., 2025), Molmo 7B (Deitke et al., 2025), JanusPro 7B (Chen et al., 2025), Qwen2.5 7B (Qwen-Team, 2024), and InternVL3 8B (Chen et al., 2024). These models were selected to represent diverse training methodologies and architectural approaches rather than to provide exhaustive coverage of all available VLMs.

Our selection encompasses a diverse set of models to explore architectural influences on color naming behavior. JanusPro uses a unified multimodal design; Molmo is optimized for rich image captioning; GLM4.1V integrates SigLIP with tailored cross-modal fusion; Qwen2.5 advances multilingual (Chinese-English) modeling; InternVL3 applies progressive training for hierarchical vision understanding; and MiniCPM-V 4.5 deliver strong vision-language and video understanding with efficient parameter use. This variety enables us to assess whether color naming reflects model-specific traits or converges across architectures. As the first systematic study of color naming in VLMs, our goal is to uncover fundamental patterns rather than rank models. All selected models fall within a similar parameter range (7–9B), ensuring fair comparison while minimizing scale-related confounds.

Additionally, we conduct an ablation study examining intra-family parameter scaling effects by comparing models of different sizes within the same architectural family (see Section 7.2 and Appendix E). This analysis helps distinguish whether observed color naming patterns stem from training methodologies and architectural choices, or simply model capacity.

### 3.3 Methodology

Following the open-ended methodology established in seminal color naming studies of Berlin & Kay (1991), we present each of the 957 color Color theaurus chips and prompt the models with "*What would you call this color?*" using additional rules to avoid verbose responses (see Appendix A for the complete prompt).

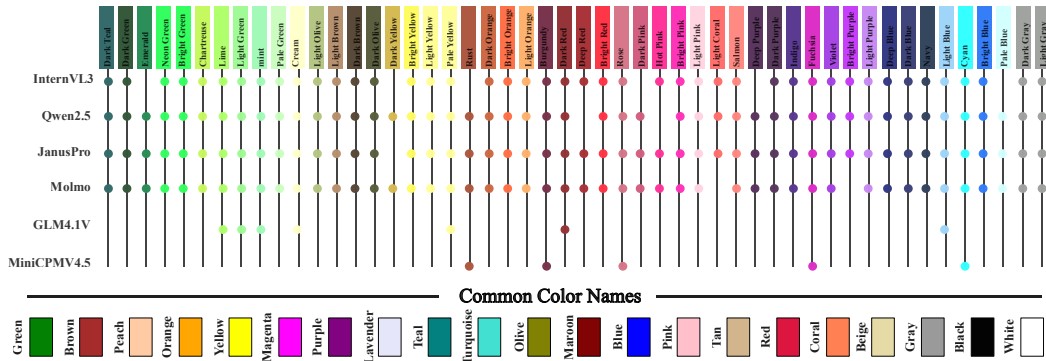

Figure 2: Common and shared color vocabularies across VLMs. **Bottom**: The 21 common terms consistently used by all five models, including basic terms like green, blue, purple, yellow, and pink, which collectively account for 67.7% of all naming responses across models. **Top**: UpSet plot showing the distribution of non-common color terms shared between subsets of models. The intersections reveal which additional color names are used by specific model combinations, highlighting vocabulary overlap patterns beyond the common core set.

Note that we employed a free-response paradigm that contrasts with usual closed-set evaluations in VLMs. The reasons are twofold: first, as stated before classic color naming experiments in humans follow the same methodology (not impose color names, but rather find them in the open ended questios). Second, that allows models to express their natural color terminology without constraining responses to predefined categories. This approach enables discovery of the full spectrum of color names that VLMs employ, from basic color terms to complex descriptive phrases.

To capture the stochastic nature of VLM responses and ensure robust sampling of each model's color naming behavior, we collected 100 independent responses per color sample using different random seeds.

Each color sample was presented as a plain RGB image (512×512 pixels) displaying the uniform color value from the dataset. Images contained no additional visual elements, text, or contextual cues that might influence color naming decisions. All models were evaluated using their default vision processing pipelines with default model temperature to encourage natural response variation while maintaining coherent outputs.

# 4    COLOR NAMES BEYOND BASIC CATEGORIES

We analyzed the distribution of color terms to identify patterns of convergence and divergence in color vocabularies. Our analysis proceeds hierarchically, beginning with common terms shared across all models before examining model-specific variations.

**Common Color Terms Across Architectures.** Despite diverse training methodologies and architectural approaches, all five models consistently employ a core set of 21 common color terms (Figure 2, bottom). These shared terms—including *green*, *blue*, *purple*, *yellow*, and *pink*—account for 67.7% of all naming responses across models. This convergence suggests these terms represent perceptual categories that emerge naturally from vision-language training rather than being explicitly programmed, echoing findings from cross-linguistic studies of human color naming (Berlin & Kay, 1991).

**Model-Specific Vocabulary Distributions.** While all models share this common core, they exhibit differences in vocabulary specificity. As shown in Figure 3, GLM4.1V and MiniCPM demonstrate highly constrained color vocabularies, with more than 88% of their responses corresponding to the 21 shared terms. In contrast, Qwen2.5 and JanusPro employ substantially more diverse color vocabularies, with roughly 50% of their responses using terms beyond the common set.

**Modifier Usage vs. Lexical Diversity.** To understand the nature of this vocabulary expansion, we analyzed whether models achieve specificity through modifiers (e.g., *light blue*, *dark red*) or through distinct lexical items (e.g., *crimson*, *turquoise*). We find evidence that models like Qwen2.5, Molmo, InternVL, and JanusPro expand basic color terms by systematically applying modifiers related to brightness, saturation, and hue (Figure 2, top. See Appendix B for more details).

This distinction has important perceptual implications. As illustrated in Figure 1, where GLM4.1V classifies a wide range of blue variations simply as *blue*, Qwen2.5 demonstrates the ability to make finer distinctions, using terms like *light blue*, *dark blue*, and *blue* to capture perceptual nuances within the blue spectrum.

**Perceptual Drivers of Color Naming** To understand which visual features guide color categorization across different vocabulary strategies, we quantify how each HSV component (Hue, Saturation, Value) contributes to color naming decisions. Using the three previous groups three groups: common colors (terms used by all models), colors with modifiers (e.g., *light blue*, *dark red*), and non-common colors without modifiers. Within each category, we discretize HSV values into bins (20 bins for Hue spanning 0-360°, 10 bins each for Saturation and Value spanning 0-100%), and encode color names using label encoding. We then calculate the mutual information score between each HSV component and the encoded color names using sklearn's *mutual_info_score*. The values in Table 2 represent the percentage contribution of each HSV component to the total mutual information ($H_{MI}$ + $S_{MI}$ + $V_{MI}$) within each category, indicating the relative importance of hue, saturation, and brightness information for different types of color naming patterns across VLMs.

Table 2 reveals systematic patterns across the three vocabulary categories. For common colors, hue dominates across all models (57-74% of total mutual information), confirming that basic color categories are primarily organized around chromatic distinctions. However, Qwen2.5 shows notably greater reliance on value (brightness) at 31% compared to other models (13-16%), suggesting a distinct perceptual weighting strategy even for common color terms. When models employ modifiers, the perceptual landscape shifts. While hue remains dominant (50-53%), both value and saturation components gain importance, reflecting the increased discriminative demands of fine-grained color naming. Qwen2.5 continues its value-centric approach with 33% contribution from brightness information, while GLM4.1V shows the highest saturation weighting (27%) among models with sufficient modifier usage. For non-common colors, hue generally maintains dominance (50-60%), but individual model strategies diverge more sharply. GLM4.1V emphasizes a relatively balanced weighting between saturation (31%) and hue (50%), while MiniCPM exhibits the most even distribution, with nearly equal reliance on hue (36%) and saturation (36%)—a unique pattern suggesting this model's non-common color decisions are driven equally by color purity and chromaticity rather than by brightness distinction. These divergent strategies indicate that vocabulary expansion beyond the common modifier framework involves model-specific perceptual mappings shaped by distinct training methodologies.

---

**Key Findings**

- All VLMs agree on 21 basic color terms despite diverse training methodologies (constrained vocabularies).
- Modifiers to the constrained set are mostly introduced as lightness modifiers rather than distinct lexical alternatives (e.g. *crimson*, *turquoise*).
- Hue consistently dominates color naming decisions across all models, but its relative importance decreases as vocabulary complexity increases—dropping for modified and non-common colors as saturation and brightness gain discriminative relevance.

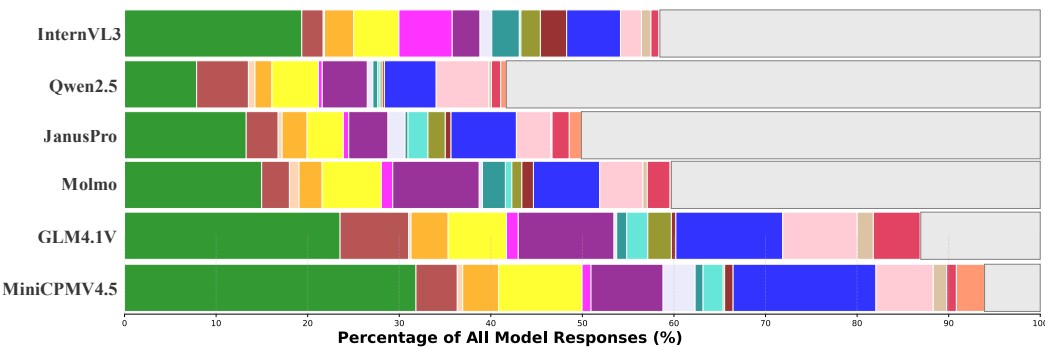

Figure 3: Distribution of common versus non-common color terms across VLMs. Each bar shows the proportion of responses using the 21 common terms (shared across all models) versus model-specific terms. GLM4.1V and MiniCPM display highly constrained vocabularies with over 88% of responses using common terms, while others exhibit greater vocabulary diversity with less than 60% of responses employing terms beyond the common set.

Table 2: Mutual Information Analysis: HSV Contributions by Color Category (%). Values represent the percentage of total mutual information contributed by each HSV component. Mutual information measures how much each component contributes to distinguishing color categories. Dashes (–) indicate insufficient data for reliable MI calculation.

| | Model | InternVL3 | Qwen2.5 | JanusPro | Molmo | GLM4.1V | MiniCPM-V-4.5 |
|---|---|---|---|---|---|---|---|
| Common Colors | Hue | **70.5** | **57.8** | **72.2** | **74.2** | **72.3** | **72.7** |
| | Saturation | 13.6 | 11.7 | 13.4 | 10.9 | 12.2 | 13.6 |
| | Value | 15.8 | 30.5 | 14.4 | 14.9 | 15.5 | 13.8 |
| Colors with Modifiers | Hue | **50.9** | **52.6** | **52.4** | **53.3** | **53.4** | – |
| | Saturation | 24.9 | 14.6 | 24.1 | 22.8 | 27.2 | – |
| | Value | 24.3 | 32.7 | 23.5 | 23.9 | 19.3 | – |
| Non-common Colors | Hue | **57.9** | **55.2** | **59.8** | **59.1** | **50.3** | **36.1** |
| | Saturation | 14.0 | 19.7 | 15.5 | 17.5 | 30.7 | **36.0** |
| | Value | 28.0 | 25.1 | 24.8 | 23.5 | 19.0 | 27.9 |

## 5 How consistent are VLMs in color naming?

Having established the vocabulary diversity across models, we now examine the consistency with which VLMs apply these color terms. This analysis addresses three fundamental questions about color naming reliability: How similar are the colors that receive identical names within each model? Which colors serve as clear prototypical examples (foci) that models name with high confidence?

### 5.1 Color Consistency

To measure color naming consistency, we employ a voting-based methodology that captures both the dominant color term assignments and their perceptual coherence. Since for each of the 957 color chips we collect 100 independent responses from each model, we assign each chip to the color name that receives the majority vote, a common practive in classic color naming (Sturges & Whitfield, 1995)) works. We then calculate consistency by measuring pairwise distances between all chips assigned to the same color name within each model's color space representation. Specifically, we convert chip RGB values to HSV coordinates and compute the mean pairwise distance for each HSV component across all chips sharing the same color label.

Figure 4 presents mean hue distance consistency across models, focusing on this HSV component as it provides the strongest explanatory power for color naming decisions. The analysis displays only 9 colors from the original 21 common terms, as we restrict the visualization

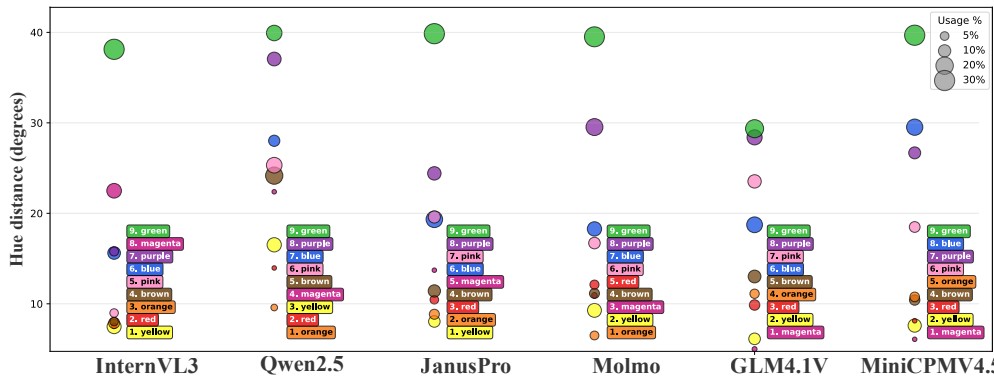

Figure 4: Hue consistency in common colors (less is more consistent), bubble size indicates usage frequency.

to colors with at least 2 assigned chips per model through majority voting, while excluding achromatic colors (black, white, gray) that lack meaningful hue information. Green consistently exhibits the largest hue distance between assigned chips across all models, which aligns with expectations given that it serves as the most frequently used color name in the common set. Conversely, yellow and orange demonstrate consistently tight hue clustering across all models, indicating more precise categorical boundaries for these color terms. Among the models, Qwen2.5 shows the lowest overall consistency in the common color set, with systematically higher hue distances across most color categories, suggesting that its expanded vocabulary comes at the cost of less precise application of basic color terms.

## 5.2 FOCAL COLORS (FOCI)

Color foci represent the most prototypical examples of each color category—the chips that consistently evoke the same color name across multiple naming trials (Berlin & Kay, 1991). In the context of VLMs, we measure foci by analyzing the stability of color term assignments across repeated responses. As before, for each of the 957 color chips, we collect 100 independent naming responses and calculate the proportion stability as the fraction of trials that produced the most common color term for that chip. A chip qualifies as a foci for a particular color name when its stability meets or exceeds a specified threshold (ranging from 0.5 to 1.0). Table 3 presents two complementary metrics across different stability thresholds: the number of unique color categories that achieve at least one foci (indicating vocabulary breadth), and the mean hue distance in degrees between all chips sharing the same color label (measuring the tightness of color clustering, where lower values indicate more precise categorical boundaries).

The results in Table 3 show that GLM4.1V and MiniCPM have the most consistent foci, maintaining 24-30 unique color categories even at the strictest threshold (1.0), while Qwen2.5 shows the most dramatic decline, dropping from 14 foci at threshold 0.5 to only 2 at threshold 1.0. This pattern aligns with our earlier finding that Qwen2.5 employs a more diverse but

Table 3: Foci Analysis: Number of Foci and Mean Hue Distance by Model and Threshold

| Model | 0.5 | | 0.6 | | 0.7 | | 0.8 | | 0.9 | | 1.0 | |
|---|---|---|---|---|---|---|---|---|---|---|---|---|
| | Foci | Dist | Foci | Dist | Foci | Dist | Foci | Dist | Foci | Dist | Foci | Dist |
| GLM4.1V | 30 | 17.4 | 29 | 17.9 | 27 | 17.2 | 25 | 16.9 | 24 | 17.4 | 24 | 17.4 |
| InternVL3 | 48 | 19.5 | 45 | 20.0 | 45 | 19.6 | 39 | 17.0 | 34 | 17.3 | 29 | 17.9 |
| JanusPro | 37 | 13.5 | 26 | 13.7 | 22 | 13.9 | 16 | 19.2 | 8 | 2.4 | 2 | 9.5 |
| MiniCPM-V-4.5 | 24 | 18.5 | 23 | 19.3 | 22 | 19.3 | 22 | 18.1 | 20 | 19.0 | 20 | 18.3 |
| Molmo | 34 | 19.2 | 29 | 18.7 | 19 | 21.0 | 15 | 13.0 | 12 | 10.6 | 8 | 5.4 |
| Qwen2.5 | 14 | 31.2 | 13 | 21.5 | 12 | 22.2 | 9 | 21.8 | 5 | 12.1 | 2 | 0.0 |

less consistent vocabulary. Regarding clustering quality, most models maintain relatively stable hue distances (15-20) across thresholds, suggesting that their color categories have consistent internal coherence regardless of strictness level. However, Qwen2.5 again stands out with notably higher hue distances at lower thresholds (31.2° at 0.5), indicating that its expanded vocabulary comes at the cost of less precise color boundaries.

---

**Key Findings**

- Vocabulary diversity comes at a consistency cost: Models achieving greater descriptive specificity sacrifice categorical precision.
- Yellow and orange show the tightest categorical boundaries in all models, while green has the largest within-category variance due to its frequent use.
- GLM4.1V, MiniCPM, and InternVL3—to a lesser extend, have more consistent foci at the different threshold levels.

---

## 6 THE ROLE OF LANGUAGE

Color naming in humans exhibits significant cultural and linguistic variation (Lindner et al., 2012). To investigate how language influences color naming consistency in VLMs, we extended our analysis to nine additional languages present in the color thesaurus dataset: Chinese, French, German, Italian, Japanese, Korean, Portuguese, Russian, and Spanish. This multilingual approach enables us to examine whether the cross-model convergence patterns observed in English generalize across linguistic boundaries, or whether language-specific factors introduce systematic variations in VLM color naming behavior.

Following our previous methodology, we prompted the selected VLMs with the same prompt as Section 3.3, translating it to each language (see supplementary material for each translation). We identified "common colors" for each language—those color terms used by all models within that language. Table 4 presents these shared vocabularies, revealing the core color terms that emerge across different VLM architectures when operating in each respective language.

The distribution of common color terms across languages reveals imbalances that likely reflect training data disparities rather than inherent linguistic differences. Chinese (22 terms) and English (21 terms) dominate the vocabulary space, while all other languages cluster substantially lower, with most maintaining fewer than half the color vocabulary available in the dominant languages. This disparity becomes particularly evident when examining languages from shared linguistic families, where expected similarities are conspicuously absent. Romance languages show surprising variation, with Spanish maintaining 13 common terms while Italian, Portuguese, and French cluster at only 6-7 terms. This pattern suggests that shared linguistic heritage does not predict similar VLM color naming, pointing instead to training-specific factors—such as dataset composition and language representation during model development—as the primary drivers of cross-linguistic color vocabulary differences.

It is important to note that the number of common color names depends on both the number and language specialization of the selected models. Some models demonstrate greater proficiency in specific languages, and when less specialized models are included in the intersection analysis, the number of common colors decreases accordingly (see Fig. 14 in appendix).

---

**Key Findings**

- Data bias drives cross-linguistic differences: English and Chinese models use over 20 common terms, while most other languages fall below 10, suggesting disparities in training data coverage rather than linguistic limitations.
- Language family does not predict color naming similarity: Romance languages show wide internal variation (Spanish: 13 terms vs. Italian/French 6-7), indicating that linguistic lineage is not a reliable predictor.

---

## 7 Ablations

### 7.1 Color-object binding

Object recognition significantly influences color perception, both in humans and in vision-language models (VLMs), where training data can bias the association between objects and color names. To examine this effect, we conducted a controlled experiment using 3D rendered objects presented in 957 distinct colors under multiple conditions. This setup allowed us to isolate the impact of object identity on color naming. The results reveal substantial object-dependent variation in color naming and modifier usage across models. These patterns demonstrate that evaluations based solely on uniform color chips, while useful as baselines, do not fully capture VLM behavior in more naturalistic contexts. Full experimental details and results are provided in Appendix D.

### 7.2 Scaling the Language Models

To isolate the role of language modeling in color naming, we conducted a controlled experiment using the InternVL family across four scales (1B, 2B, 8B, and 14B parameters), keeping the visual encoder and training strategy constant. This design ensures that any observed differences stem from the language model component alone. The results show that language model architecture significantly affects color vocabulary usage, with notable shifts in both common color frequency and individual color preferences across scales. These findings highlight that multimodal color understanding depends not only on visual perception but also critically on the language model's ability to map visual features to linguistic categories. Full results and experimental details are provided in Appendix E.

## 8 Related Work

Recent research on color understanding in vision systems has taken several approaches. Alabau-Bosque et al. (Alabau-Bosque et al., 2025) explored human-model color-word alignment using a gamified concept-to-color CLIP mapping. Other works, such as Akbarinia et al. (Akbarinia, 2025) and Arias et al. (Arias et al., 2024), investigate internal color representations through linear probes and mechanistic analysis of models like CLIP, focusing on how networks encode color categories rather than linguistic output. Other work evaluates color robustness using Ishihara color blindness tests (Samin et al., 2024; Ye et al., 2025; Ling et al., 2025; Hayashi et al., 2025) or examines color-language associations through prompted language models with hexadecimal codes (Mukherjee et al., 2024). ColorBench (Liang et al., 2025) provides the most comprehensive evaluation of VLM color capabilities across 1,400+ instances spanning 11 task types, but employs multiple-choice formats that constrain responses to predefined categories. In contrast, our work examines the natural color vocabularies that emerge from VLMs through unconstrained naming tasks, replicating classic psycholinguistic methodologies to understand what color terms VLMs actually use and how naming patterns vary across architectures, languages, and contexts.

## 9 Conclusions

In this work, we perform a systematic evaluation of color naming across vision-language models. Our analysis reveals that while VLMs align well with human naming for prototypical colors, they diverge significantly on expanded, non-prototypical color sets. Rather than the 11 basic categories found in human studies, VLMs converge on 21 common color terms through two distinct strategies: constrained models using only core terms versus expansive models employing lightness modifiers for finer discrimination. Cross-linguistic and object-specific experiments reveal severe training imbalances favoring English and Chinese, alongside context-sensitive color naming where identical colors receive different labels based on object identity. Finally, ablation studies demonstrate that language model architecture significantly influences color naming independent of visual processing capabilities.

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
