## A  PROMPTS USED

### A.1  PROMPT FOR BERLIN AND KAY'S EXPERIMENT

*You are participating in a color naming survey. Look at the color shown in the image and provide only the name you would naturally use to describe this color.*
*Your task is to name the color using only a single word - a monolexemic color term.*
*Rules:*
*- Use only ONE word*
*- Do not use compound terms (no "blue-green", "red-orange", "yellow-green")*
*- Do not use modifiers (no "yellowish", "light", "dark", "pale", "bright", "deep")*
*- Do not use descriptive phrases or multiple words*
*- Choose the most basic, common color term that best describes what you see*
*- Do not explain your choice or provide additional commentary*
*What would you call this color?*

### A.2  PROMPT FOR MAIN EXPERIMENT

*You are participating in a color naming survey. Look at the color shown in the image and provide only the name you would naturally use to describe this color.*
*Rules:*
*- Give only a simple color name or color description*
*- Use everyday language*
*- Do not provide explanations, hex codes, or technical details*
*- Do not say "This color appears to be..." or similar phrases*
*- Just state the color name directly*
*What would you call this color?*

### A.3  PROMPTS FOR LANGUAGE EXPERIMENT

#### A.3.1  CHINESE

您正在参与一项颜色命名调查。请看图像中显示的颜色,
仅提供您会自然使用的颜色名称来描述这种颜色。
规则:
- 只提供简单的颜色名称或颜色描述
- 使用日常用语
- 不要提供解释、十六进制代码或技术细节
- 不要说"这种颜色看起来像..."或类似的短语
- 直接说出颜色名称
您会如何称呼这种颜色?

#### A.3.2  FRENCH

*Vous participez à une enquête sur la dénomination des couleurs. Regardez la couleur montrée dans l'image et fournissez uniquement le nom que vous utiliseriez naturellement pour décrire cette couleur.*
*Règles :*
*- Donnez seulement un nom de couleur simple ou une description de couleur*
*- Utilisez un langage quotidien*
*- Ne fournissez pas d'explications, de codes hexadécimaux ou de détails techniques*
*- Ne dites pas "Cette couleur semble être..." ou des phrases similaires*

*- Énoncez simplement le nom de la couleur directement*
*Comment appelleriez-vous cette couleur ?*

### A.3.3 GERMAN

*Sie nehmen an einer Farbnamenumfrage teil. Schauen Sie sich die im Bild gezeigte Farbe an und geben Sie nur den Namen an, den Sie natürlicherweise verwenden würden, um diese Farbe zu beschreiben.*
*Regeln:*
*- Geben Sie nur einen einfachen Farbnamen oder eine Farbbeschreibung an*
*- Verwenden Sie Alltagssprache*
*- Geben Sie keine Erklärungen, Hex-Codes oder technische Details an*
*- Sagen Sie nicht "Diese Farbe scheint zu sein..." oder ähnliche Phrasen*
*- Nennen Sie einfach direkt den Farbnamen*
*Wie würden Sie diese Farbe nennen?*

### A.3.4 ITALIAN

*Stai partecipando a un'indagine sulla denominazione dei colori. Guarda il colore mostrato nell'immagine e fornisci solo il nome che useresti naturalmente per descrivere questo colore.*
*Regole:*
*- Fornisci solo un nome di colore semplice o una descrizione del colore*
*- Usa un linguaggio quotidiano*
*- Non fornire spiegazioni, codici esadecimali o dettagli tecnici*
*- Non dire "Questo colore sembra essere..." o frasi simili*
*- Dichiara semplicemente il nome del colore direttamente*
*Come chiameresti questo colore?*

### A.3.5 JAPANESE

あなたは色の命名調査に参加しています。画像に表示された色を見て、この色を表現するために自然に使う名前のみを提供してください。
ルール：
- シンプルな色の名前または色の説明のみを提供する
- 日常的な言葉を使用する
- 説明、16進コード、または技術的な詳細は提供しない
- 「この色は...のように見えます」または類似のフレーズは言わない
- 色の名前を直接述べる
この色を何と呼びますか？

### A.3.6 KOREAN

당신은 색상 명명 조사에 참여하고 있습니다. 이미지에 표시된 색상을 보고 이 색상을 표현하기 위해 자연스럽게 사용할 이름만을 제공해 주세요.
규칙:
- 간단한 색상 이름이나 색상 설명만 제공하세요
- 일상적인 언어를 사용하세요
- 설명, 16진수 코드, 또는 기술적 세부사항을 제공하지 마세요
- "이 색상은 ...처럼 보입니다" 또는 유사한 표현을 사용하지 마세요
- 색상 이름을 직접 말하세요
이 색상을 무엇이라고 부르시겠습니까?

### A.3.7 PORTUGUESE

*Você está participando de uma pesquisa sobre nomenclatura de cores. Olhe para a cor mostrada na imagem e forneça apenas o nome que você usaria naturalmente para descrever esta cor.*
*Regras:*
*- Forneça apenas um nome de cor simples ou descrição de cor*
*- Use linguagem cotidiana*

*- Não forneça explicações, códigos hexadecimais ou detalhes técnicos*
*- Não diga "Esta cor parece ser..." ou frases similares*
*- Apenas declare o nome da cor diretamente*
*Como você chamaria esta cor?*

### A.3.8 Russian

*Вы участвуете в опросе по именованию цветов. Посмотрите на цвет, показанный на изображении, и предоставьте только название, которое вы бы естественно использовали для описания этого цвета.*
*Правила:*
*- Дайте только простое название цвета или описание цвета*
*- Используйте повседневный язык*
*- Не предоставляйте объяснения, шестнадцатеричные коды или технические детали*
*- Не говорите "Этот цвет выглядит как..." или похожие фразы*
*- Просто назовите цвет напрямую*
*Как бы вы назвали этот цвет?*

### A.3.9 Spanish

*Estás participando en una encuesta sobre nomenclatura de colores. Mira el color mostrado en la imagen y proporciona solo el nombre que usarías naturalmente para describir este color.*
*Reglas:*
*- Da solo un nombre de color simple o descripción de color*
*- Usa lenguaje cotidiano*
*- No proporciones explicaciones, códigos hexadecimales o detalles técnicos*
*- No digas "Este color parece ser..." o frases similares*
*- Solo declara el nombre del color directamente*
*¿Cómo llamarías a este color?*

## B  Color Modifiers

Figure 5 reveals that most non-common color terms employed by Qwen2.5, Molmo, InternVL and JanusPro consist of basic color terms with modifiers rather than entirely different color names. This indicates that these models develop more fine-grained color categories by systematically applying brightness, saturation, and hue modifiers to the common set.

## C  Common Color Names by Language

Table 4 presents the complete set of common color terms identified across all evaluated VLMs for each language. These terms represent the intersection of color vocabularies—those color names consistently used by all models when operating in a specific language. The variation in vocabulary sizes across languages reinforces our finding of substantial training data imbalances in current VLMs. Chinese maintains the largest common vocabulary with 22 terms, followed closely by English with 21 terms, while most other languages demonstrate significantly constrained vocabularies ranging from 6-13 terms. Notably, the color terms are presented with their corresponding representative colors from the dataset, illustrating the perceptual mappings that VLMs have learned for each linguistic category. The reduction in vocabulary for non-dominant languages suggests that current VLM training procedures have not achieved equitable representation across linguistic communities, with important implications for multilingual deployment of vision-language systems in color-sensitive applications.

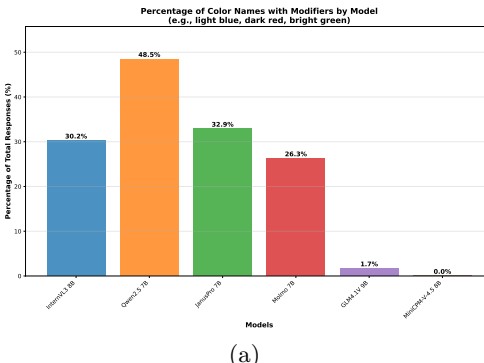 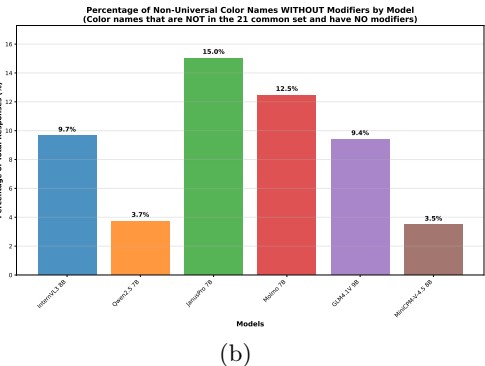

(a)                                    (b)

Figure 5: Analysis of color vocabulary expansion strategies across VLMs. **Left**: Percentage of color terms using modifiers (e.g., *light blue, dark red*) by model. **Right**: Percentage of color terms that are both non-common and non-modifier based (i.e., distinct lexical items like *crimson, turquoise*). Results demonstrate that models with expanded vocabularies (Qwen2.5, Molmo, InternVL3, JanusPro) achieve color specificity primarily through systematic application of modifiers to basic color terms rather than employing entirely different color names, while constrained models (GLM4.1V, MiniCPM) rely predominantly on unmodified basic terms.

Table 5 reveals patterns in color naming frequency distributions across languages. Green dominates as the most frequently used color term in seven out of ten languages, with particularly high usage in Romance languages (Italian 62.5%, Spanish 62.2%, Portuguese 58.1%) and Japanese (61.9%). However, Russian and Korean diverge from this pattern, with blue serving as their most frequent color term (51.2% and 52.4% respectively). The variation in frequency percentages—green ranging from 30.8% in Russian to 62.5% in Italian—suggests that language-specific factors beyond simple perceptual commons influence color naming distributions.

Table 4: Common colors by language

| Language | Common Colors | Count |
|---|---|---|
| Chinese | 亮绿色, 棕色, 橄榄绿, 橙色, 浅棕色, 浅蓝色, 淡紫色, 深棕色, 深红色, 深绿色, 深蓝色, 灰色, 白色, 米色, 粉红色, 紫色, 红色, 绿色, 蓝绿色, 蓝色, 青色, 黑色 | 22 |
| English | green, brown, peach, orange, black, yellow, magenta, purple, lavender, teal, turquoise, olive, maroon, blue, pink, gray, tan, red, coral, beige, white | 21 |
| Spanish | amarillo, azul, azul claro, azul oscuro, blanco, coral, morado, rojo, rosa, verde, verde claro, verde oliva, verde oscuro | 13 |
| German | blau, braun, gelb, grau, lila, magenta, orange, pink, rot | 9 |
| Russian | зеленый, коричневый, красный, лиловый, лимонный, оранжевый, синий, фиолетовый | 8 |
| Italian | arancione, azzurro, grigio, grigio chiaro, marrone, verde acqua, violetto | 7 |
| Portuguese | azul-marinho, laranja, rosa claro, verde claro, vermelho, vermelho escuro, violeta | 7 |
| French | bleu, gris, marron, rose, rouge, violet | 6 |
| Japanese | ピンク, 水色, 紫, 緑, 赤, 青 | 6 |
| Korean | 노란색, 녹색, 보라색, 파란색, 회색, 흰색 | 6 |

Table 5: Average Color Mention Frequencies by Language (Top 5 Colors) thesaurus

| Language | 1st Most Frequent | 2nd Most Frequent | 3rd Most Frequent | 4th Most Frequent | 5th Most Frequent |
|---|---|---|---|---|---|
| English | Green (38.2%) | Blue (22.3%) | Purple (17.8%) | Yellow (16.9%) | Orange (11.4%) |
| Chinese | Green (48.9%) | Blue (25.7%) | Red (22.0%) | Yellow (18.8%) | Orange (15.8%) |
| French | Green (59.3%) | Blue (51.2%) | Yellow (41.8%) | Red (31.7%) | Orange (24.8%) |
| German | Blue (54.3%) | Green (52.9%) | Red (46.4%) | Yellow (40.7%) | Orange (32.6%) |
| Italian | Green (62.5%) | Orange (42.4%) | Blue (41.8%) | Yellow (38.1%) | Red (25.9%) |
| Japanese | Green (61.9%) | Blue (50.7%) | Orange (39.5%) | Red (31.5%) | Yellow (29.4%) |
| Korean | Blue (52.4%) | Yellow (40.5%) | Green (38.9%) | Red (27.7%) | Orange (26.6%) |
| Portuguese | Green (58.1%) | Blue (50.5%) | Orange (45.9%) | Red (35.1%) | Yellow (26.4%) |
| Russian | Blue (51.2%) | Red (39.9%) | Green (30.8%) | Orange (28.8%) | Yellow (17.8%) |
| Spanish | Green (62.2%) | Blue (51.4%) | Orange (37.1%) | Yellow (35.2%) | Red (27.4%) |

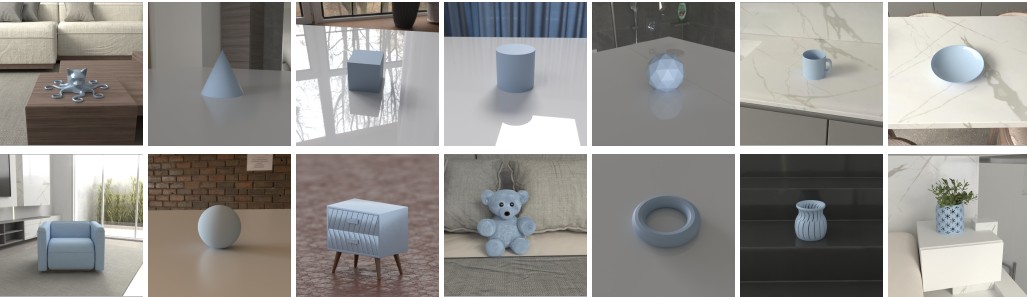

Figure 6: 3D rendered objects used in the color-object binding experiment. The stimulus set includes geometric primitives (cone, cube, cylinder, icosphere, sphere, torus) and everyday objects (cat sculpture, mug, plate, sofa, table, teddybear, vase). Each object was systematically rendered in all 957 colors from the Color Thesaurus dataset.

# D  COLOR-OBJECT BINDING - DETAILS

As stated earlier, real-world color perception is significantly complicated by object-color interactions. Research in human color perception has established that object recognition fundamentally alters color appearance through memory color effects, an effect that is further exacerbated in VLMs where training datasets can induce bias in the color names that models associate with certain objects. To investigate this phenomenon, we conducted a controlled experiment using 3D rendered objects where each object could be systematically presented in all 957 colors from the Color Thesaurus dataset under multiple illumination and scene conditions. Our object set included both geometric primitives (cone, cube, cylinder, icosphere, sphere, torus) and everyday items (cat sculpture, mug, plate, sofa, table, teddybear, vase), as shown in Figure 6. This approach allows us to isolate the influence of object identity on color naming decisions while spanning diverse object categories that might exhibit different color-naming biases.

As Figures 7, 8, and 9 demonstrate, models exhibit significant shifts in color name distribution depending on the object being evaluated. For example, Qwen2.5 shifts from using 55% common colors when evaluating color chips to 90% when evaluating objects. This object-dependent variation extends to modifier usage patterns. Figure 10 shows that InternVL3 uses color names with modifiers for approximately 54% of responses when describing sofas, but this percentage decreases dramatically to 6.4% for cubes. A similar analysis can be extracted for Qwen2.5 (see Figure 11), while as it was the case in other experiments, MiniCPM-V-4.5 behaves in quite a different manner (see Figure 12).

These object-dependent variations reveal that the controlled color naming patterns observed with uniform color chips represent an essential but incomplete picture of VLM color behavior in naturalistic settings. Rather than invalidating chip-based analysis, these findings underscore its critical value as a controlled baseline for understanding VLM color naming competencies independent of contextual biases.

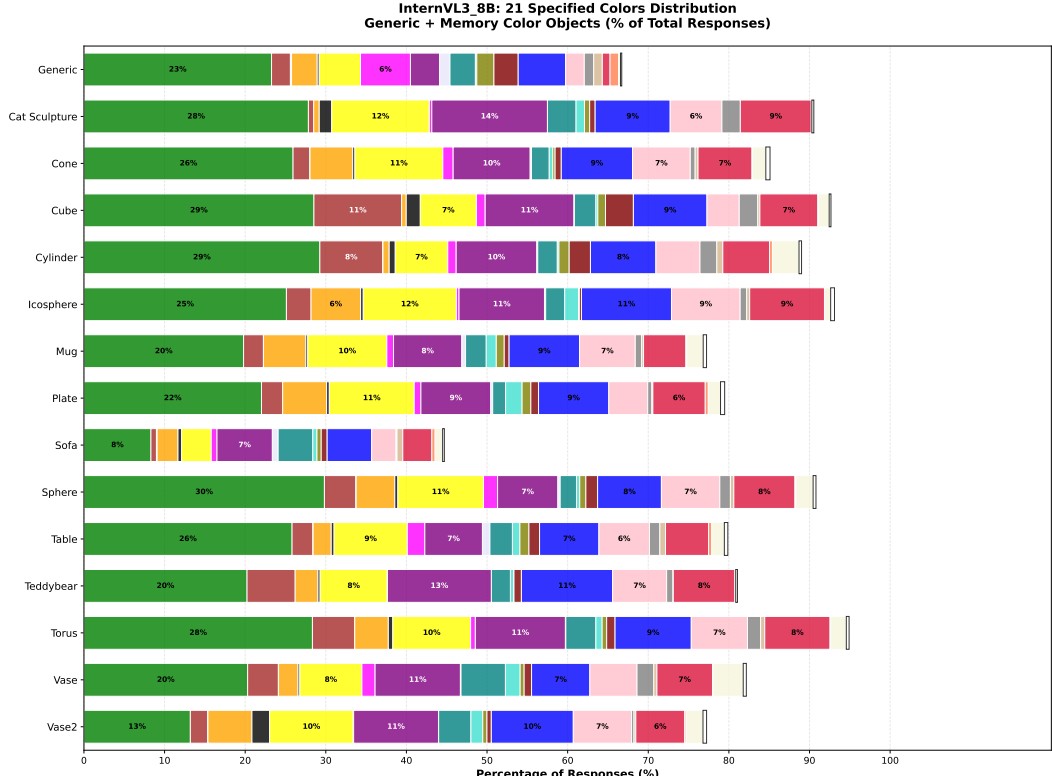

Figure 7: Object-dependent variation in common color term usage for InternVL3 8B. Stacked bars show the proportion of responses using the 21 common color terms versus model-specific terms across different object types.

## E  THE ROLE OF LANGUAGE MODELS IN COLOR NAMING -DETAILS

To isolate the specific contribution of language modeling to color naming behavior, we conducted a controlled experiment using the InternVL family across four different language model scales: 1B, 2B, 8B, and 14B parameters. This experimental design holds the visual processing pipeline constant—all variants employ the identical vision encoder (InternViT-300M-448px-V2.5) and training strategy (Zhu et al., 2025)—while varying only the language model component.

Figure 13 reveals that language model architecture significantly influences color vocabulary usage even when visual feature extraction remains identical. Common color usage fluctuates across language model scales: 73% at 1B, dropping to 60% at 8B, then increasing to 79% at 14B. Individual color frequencies also vary dramatically—brown occupies 10% of responses with the 1B language model but decreases to 3% with the 8B variant. These patterns demonstrate that color naming consistency depends not only on visual perception capabilities but critically on the language model's capacity to map visual features onto linguistic color categories, highlighting the crucial role of language architecture in multimodal color understanding.

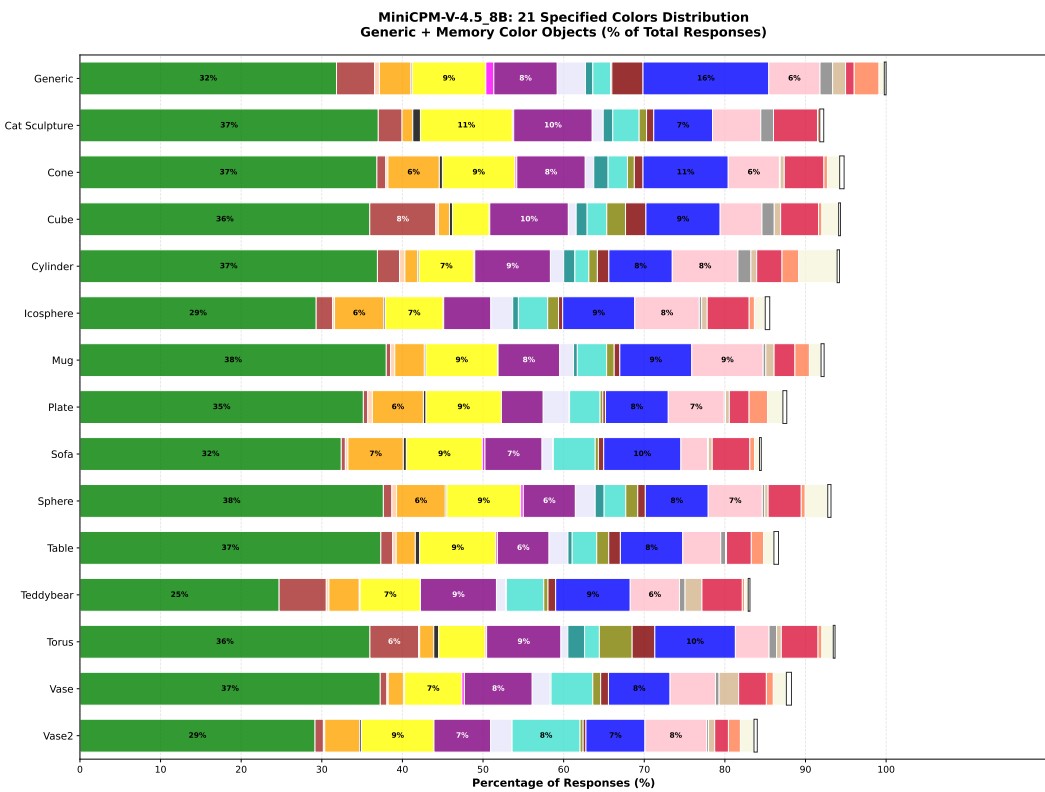

Figure 8: Object-dependent variation in common color term usage for MiniCPM-V-4.5 8B. Stacked bars show the proportion of responses using the 21 common color terms versus model-specific terms across different object types.

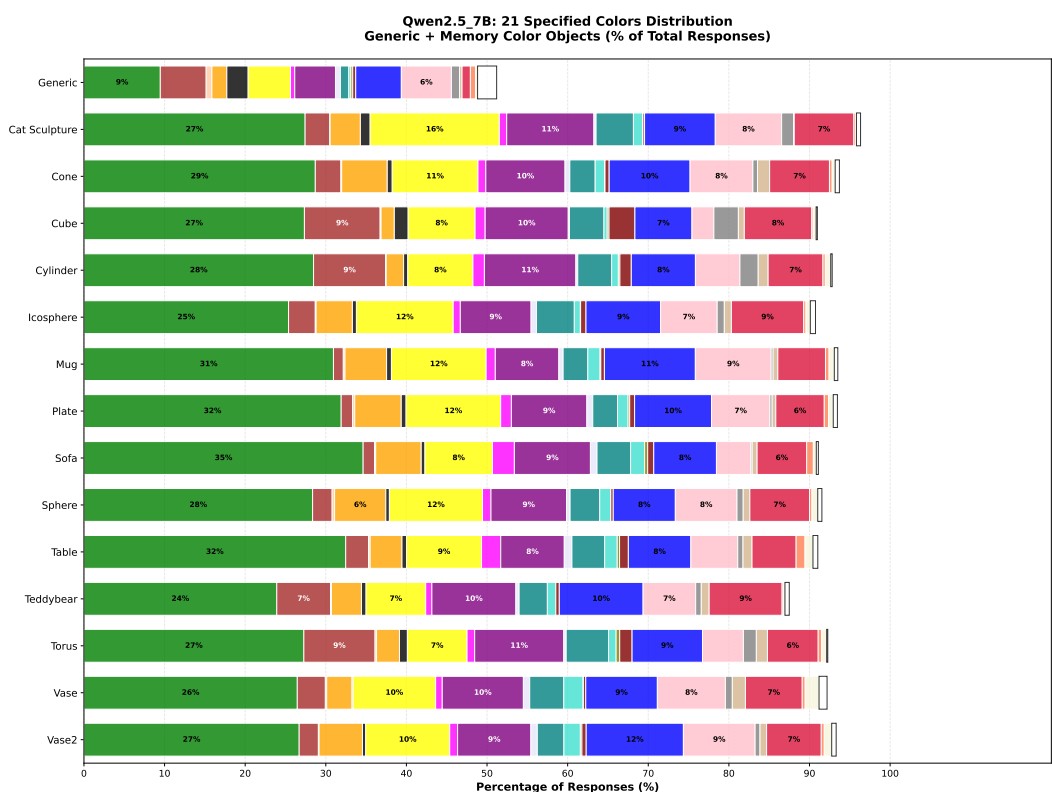

Figure 9: Object-dependent variation in common color term usage for Qwen2.5 7B 8B. Stacked bars show the proportion of responses using the 21 common color terms versus model-specific terms across different object types.

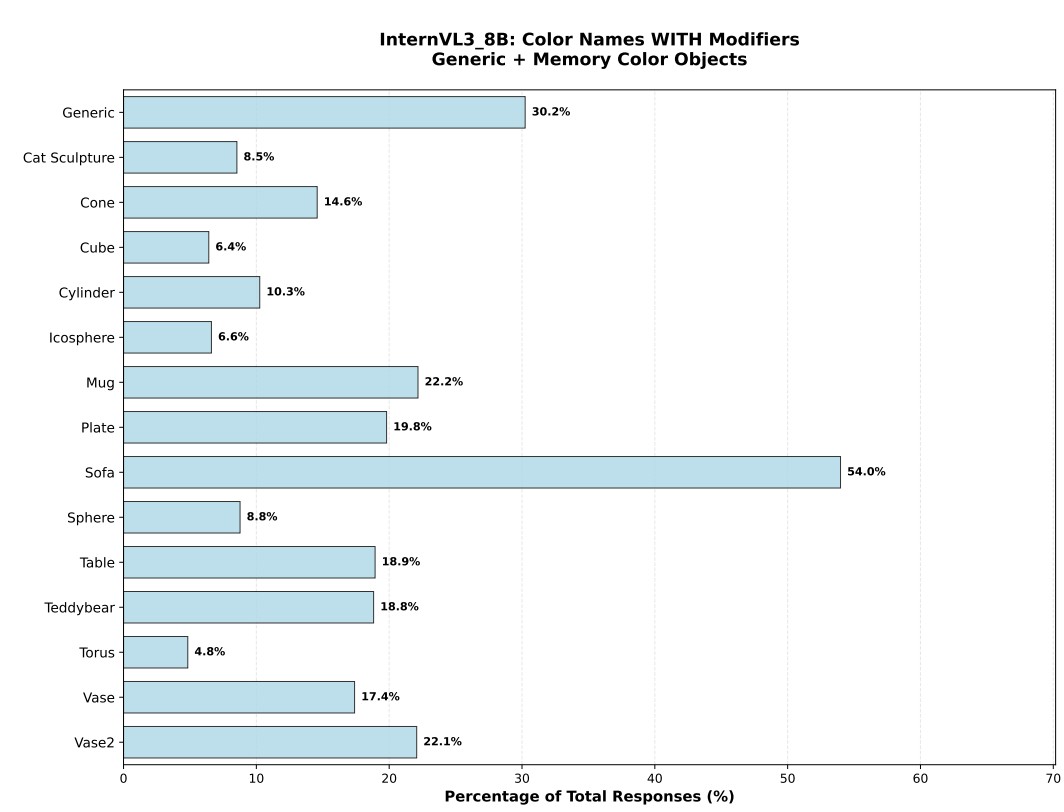

Figure 10: Modifier usage variation across object types for InternVL3 8B. Stacked bars show the proportion of color responses containing modifiers (light, dark, etc.) versus unmodified color terms for different objects.

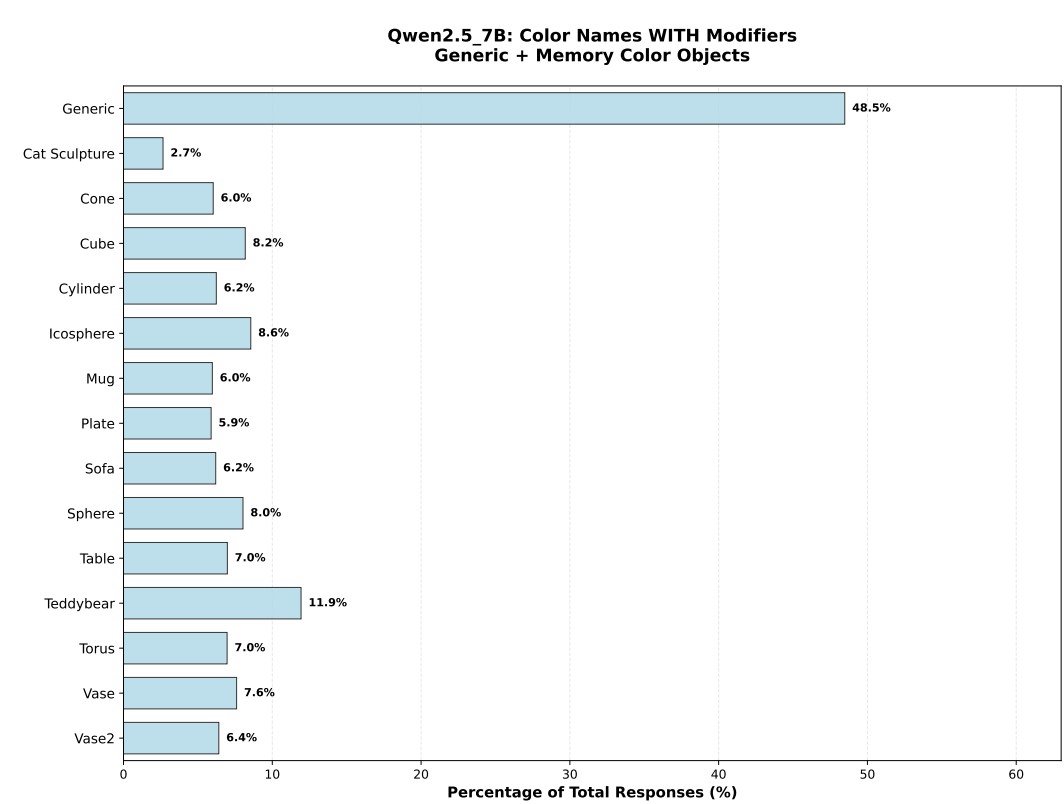

Figure 11: Modifier usage variation across object types for Qwen2.5 7B. Stacked bars show the proportion of color responses containing modifiers (light, dark, etc.) versus unmodified color terms for different objects.

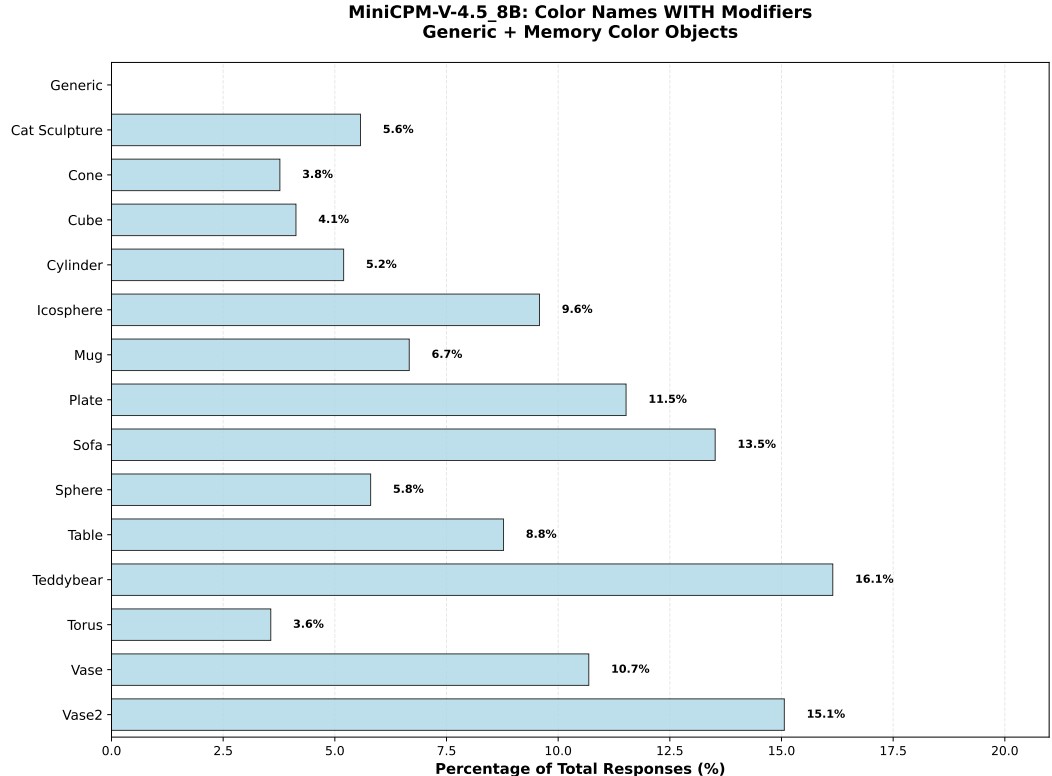

Figure 12: Modifier usage variation across object types for MiniCPM-V-4.5. Stacked bars show the proportion of color responses containing modifiers (light, dark, etc.) versus unmodified color terms for different objects.

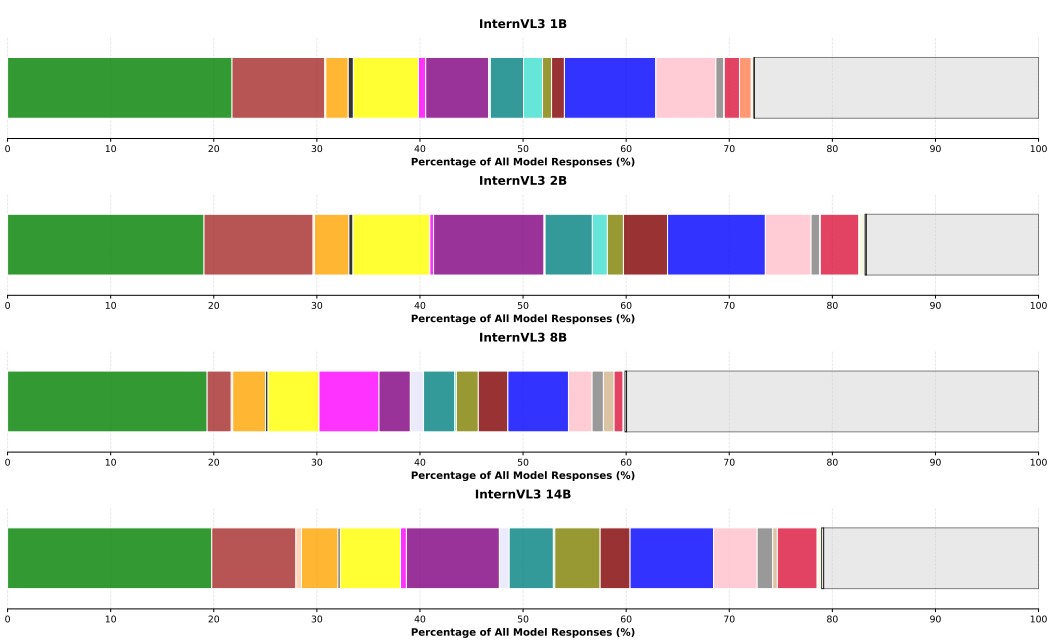

Figure 13: Language model parameter scaling effects on color vocabulary distribution within the InternVL family. Each bar shows the proportion of responses using the 21 common color terms versus model-specific terms across four different language model scales (1B, 2B, 8B, 14B parameters) while maintaining identical visual processing pipelines.

1242
1243
1244
1245
1246
1247
1248
1249
1250
1251
1252
1253
1254
1255
1256
1257
1258
1259
1260
1261
1262
1263
1264
1265
1266
1267
1268
1269
1270
1271
1272
1273
1274
1275
1276
1277
1278
1279
1280
1281
1282
1283
1284
1285
1286
1287
1288
1289
1290
1291
1292
1293
1294
1295

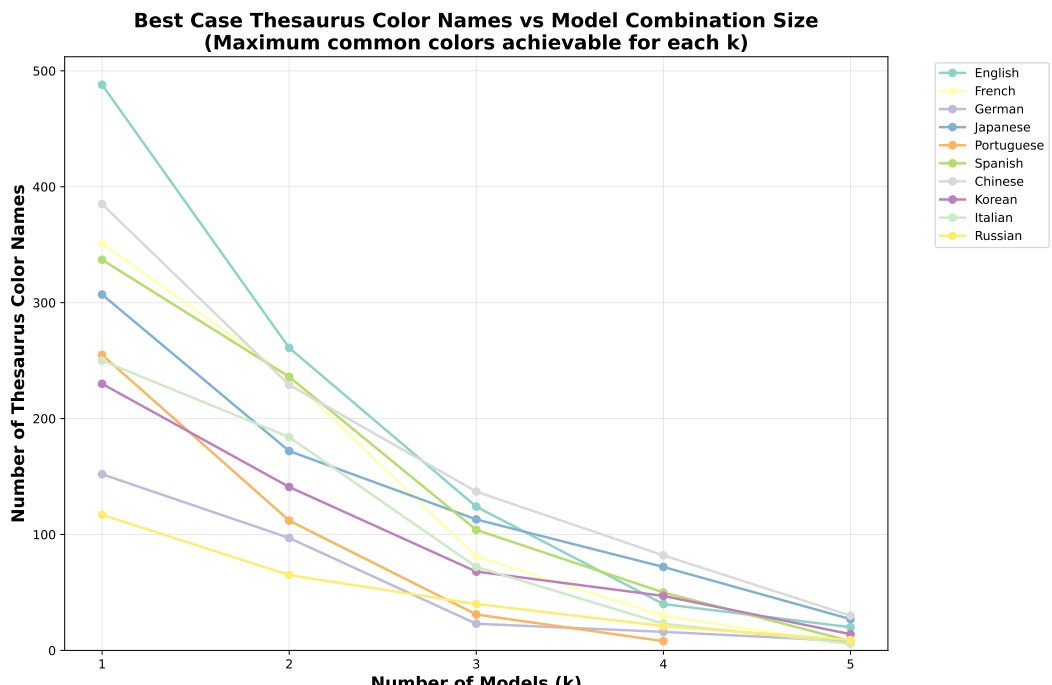

Figure 14: Cross-linguistic analysis of maximum common color vocabulary under optimal model selection. Each curve represents a different language, showing the maximum number of common color terms achievable when requiring agreement across k models (x-axis) through optimal model combination selection. Common color terms are defined as those appearing in all k selected models within each language. The y-axis indicates the maximum number of such terms obtainable by choosing the best possible subset of k models from our VLM suite.