# OpenReview forum: "Color Names in Vision-Language Models"
_ICLR.cc/2026/Conference — ICLR 2026 Conference Withdrawn Submission_

### Official Review · Reviewer_vJTM · 2025-10-19

**Soundness:** 2
**Presentation:** 2
**Contribution:** 1
**Rating:** 2
**Confidence:** 4

**Summary:**

The paper presents an analysis of how Vision-Language Models (VLMs) assign color names to a set of 957 color chips. As part of a motivational experiment on color categorization, the authors report that when using 111 chips, the models achieved nearly 90% accuracy, while performance slightly declined when the number of chips increased to 330. Through the color-naming experiments, the paper analyzes which 21 color terms are most frequently used, as well as the vocabulary distribution and lexical diversity across different models. Additionally, a consistency analysis is conducted to examine how similar colors are when assigned the same name, and how consistently each color is labeled with a single term. The study also explores cross-lingual behavior, evaluating model responses to queries posed in multiple languages.

**Strengths:**

- The paper conducts a broad range of experiments across multiple models, exploring color perception from various perspectives

**Weaknesses:**

- As ColorBench (mentioned in the Related Work) already provides extensive analyses on color perception, it would be helpful if the paper could clearly summarize what unique analytical insights or findings are newly provided by this work in comparison to ColorBench.
- [L57–58] The motivation is not entirely convincing. Humans are generally not expected to recall the exact name of a color when seeing it. For instance, even if a color is a slightly lighter shade of blue, people would still recognize it as “blue” in the absence of other blue objects. Thus, the motivation for expecting precise color naming feels less intuitive.
- In Table 1, all models achieve around 90% accuracy in categorizing 111 color chips into 11 color groups. This level of performance already seems sufficient, so including examples or scenarios where finer-grained color description is genuinely necessary would make the motivation more compelling.
- There is an inconsistency between L137 (“six representative vision-language models”) and the abstract (“five models”), which should be clarified.
- The finding in Sec. 5.1 that green has the largest hue distance has already been reported in prior work, such as [A] Hyeon-Woo et al., “VLM’s Eye Examination: Instruct and Inspect Visual Competency of Vision-Language Models,” TMLR 2025. Hence, the contribution here is somewhat limited.
- In Table 3, it would be helpful to include the ranges of Foci and Dist values for better interpretability.
- In Section 6, the conclusion that English and Chinese exhibit the highest color diversity is attributed to differences in dataset composition. However, since most models are not genuinely multilingual, this result seems rather expected and therefore less insightful.

**Questions:**

- Do the authors believe that improving a model’s ability to name colors would also have a positive impact on other vision-language tasks? What is the underlying motivation or expected benefit of conducting such an analysis?
- Since color perception is often relative rather than absolute, comparative understanding (e.g., identifying which object is more reddish or brighter) may be more meaningful in real-world scenarios. How do the authors view this aspect, and have they considered evaluating models under relative color comparison settings?

---

### Official Review · Reviewer_C3UJ · 2025-10-31

**Soundness:** 3
**Presentation:** 3
**Contribution:** 2
**Rating:** 2
**Confidence:** 4

**Summary:**

This submission presents the first systematic investigation into how Vision-Language Models (VLMs) name colors, inspired by classic psycholinguistic studies on human color naming. The authors evaluate six prominent VLMs (GLM4.1V, Qwen2.5, JanusPro, Molmo, MiniCPM-V4.5, InternVL3) using open-ended prompts (e.g., "What would you call this color?"), allowing for free-form responses beyond predefined color labels.

The study finds that while VLMs achieve near-human performance on prototypical colors, their accuracy drops significantly for desaturated or non-prototypical hues, highlighting a gap in their fine-grained color understanding. Across different architectures, the models consistently converge on 21 common color terms, which the authors analyze in depth. Two additional ablation studies explore 1) color–semantic entanglement (but already known in prior work such as [C1]), and 2) the effect of model size on naming behavior.

A cross-linguistic extension (covering nine languages) offers an interesting perspective but currently feels somewhat disconnected from the core analysis, possibly due to limited integration with the main research questions.



**Comments**

- The paper presents a novel and timely perspective that bridges cognitive linguistics and modern multimodal AI, which could be valuable for both communities. However, the practical motivation and broader implications of the findings could be clarified.

- Some of respective analyses are informative but would benefit from being tied together through a unifying application or use case. For instance, illustrating how color naming behavior might affect downstream tasks (e.g., having a useful relationship between the color names used and visual colors, such creative writing of poem from visual images) could make the study’s relevance more concrete.

- The cross-linguistic experiments are novel but could be better contextualized: perhaps by linking observed language-specific variations to the models’ training data or to known linguistic universals in color naming.

- Many claims sounds intuitively guessable. Reducing the findings and claims that may appear to be obvoius and focusing on more non-trivial and idenfiying the cause of the problem would significantly improve the message of the paper.

**Overall Evaluation**

This paper makes an interesting and original contribution by exploring an underexamined aspect of VLMs, color naming. With stronger motivation, tighter narrative integration between sections, and a clearer articulation of practical implications, the work could have substantial impact and serve as a valuable reference for future research at the intersection of vision, language, and cognition.


[C1] Mitigating Object Hallucinations in Large Vision-Language Models through Visual Contrastive Decoding, CVPR 2024.

**Strengths:**

- First comprehensive, human-style evaluation of color naming in VLMs, integrating psycholinguistics with multimodal AI evaluation.

- Introduces new metrics (color foci stability, mutual information of HSV channels) to quantify perceptual-linguistic alignment.

- Analyzes both inter-model convergence (shared vocabularies) and intra-model consistency (stability, foci).

- Includes multilingual and multimodal ablations (object binding, language scaling), offering a diverse understanding of VLM color cognition.

**Weaknesses:**

1. This reviewer feels that the multilingual analysis (Sec. 6) is not well blended with the scope of this work.
1. In Sec. 6, the first finding "Data bias drives" sounds obvious. I guess that's why the authors referred to it as a "finding" without clear evidence. To call it as a "finding", the color name distribution in the training dataset and the prediction distribution should be compared. This reviewer understands that the training datasets in most cases are not publicly available. However, to support the hypothesis, at least a small snippet would be necessary. Otherwise, it should have been presented as a hypothesis.
1. Likewise, most of the findings imply that the lagging performance is likely to be caused by training data imbalance. Then, to clearly validate the arguments, the color name distribution in the training dataset should be compared with the occurrence of the predicted color terms. Thereby, one can clearly understand how much correlation they have. This would be important, because it may suggest how to remedy the issue. The submission did not show any supporting evidence about it.
1. Key quantitative results (accuracy drops, mutual information, foci stability) lack formal significance testing or confidence intervals.
1. While ablation studies show that language model scaling affects color naming, identifying more detailed causes (e.g., embedding structure, decoder bias) remains speculative and would improve the quality of the submission.
1. The paper could further explore how its findings relate to applied use cases in practice.
1. The submission does not present the important link from the reported color test performance and the down-stream task performance to clearly demonstrate the impact of the limited performance in practice.

**Questions:**

1. While the authors presented very fine-grained prediction as a human reference in Fig. 1, this reviewer doubts that most ordinary humans are not good at this task. Then, why do we need to compare and be motivated by humans? Also, what is the implication of these findings in practical applications?

1. In Sec. 6, since the models are trained with imbalanced training data according to languages, it is obvious that minor languages in the training dataset would have less resolution in color naming. A simple yet strong baseline must be to infer in English and translate the color names to other languages with machine translation. This should produce accurate and strong results. Given this baseline, why is this section important in the context of the submission?

---

### Official Review · Reviewer_8uH4 · 2025-11-02

**Soundness:** 2
**Presentation:** 1
**Contribution:** 2
**Rating:** 2
**Confidence:** 4

**Summary:**

This paper presents the first systematic evaluation of color naming behavior in Vision-Language Models (VLMs). The authors replicate classical human color-naming methodologies using 957 color samples to test six recent VLMs, including GLM4.1V, Qwen2.5, JanusPro, Molmo, MiniCPM, and InternVL3. The paper establishes an empirical foundation for evaluating linguistic color perception in multimodal AI systems.

**Strengths:**

1. This paper presents the first systematic evaluation of color naming in vision-language models, revealing that while VLMs perform strongly on prototypical colors, their accuracy deteriorates notably on non-prototypical or ambiguous hues.

2. The experimental design is comprehensive and well-controlled, encompassing 957 color samples, 100 stochastic responses per sample, six architecturally diverse models, and a series of ablation studies addressing model scaling, object–color binding, and multilingual effects.

**Weaknesses:**

1. The paper’s motivation is not clearly articulated. While Section 2 mentions the comparison to Berlin and Kay’s color categorization experiment, it remains unclear what the overarching goal of the study is. Is the paper primarily aiming to evaluate whether VLMs replicate human color categorization patterns, or is it testing a broader hypothesis about model cognition? A more explicit statement of the research objective would help the reader understand the significance and scope of the work.

2. The paper’s structure and writing hinder comprehension. The main ideas within sections or subsections are not always clearly distinguished, making it difficult to follow the logical flow of arguments.

**Questions:**

See above

---

### Official Review · Reviewer_9hsW · 2025-11-03

**Soundness:** 2
**Presentation:** 2
**Contribution:** 1
**Rating:** 2
**Confidence:** 2

**Summary:**

This paper presents a systematic evaluation of color naming capabilities in large VLMs. The authors identify 21 common color terms shared across all models and characterize two vocabulary strategies: models using predominantly basic terms versus those employing systematic lightness modifiers. They also reveal color naming bias from training corpus with a cross-linguistic analysis.

**Strengths:**

**1. Well-motivated research question:** Understanding how VLMs name colors is practically important and interesting given their widespread deployment.

**2. Comprehensive evaluation:** The 957 color samples provide broader coverage than typical basic color category tests, and the cross-linguistic analysis adds valuable breadth and robustness.

**Weaknesses:**

**1. Limited Technical Contribution:** The paper mostly relies on prompt-based experiments with little to none statistical significance testing. Neither do the authors provide rigorous experiments nor a helpful benchmark or constructive insights.

**2. Limited Model Selection:** As an observational study, the paper only includes 5 open-source small-size VLMs. One might be even more curious about powerful commercial models like GPT, Claude, etc., given that this experimental setup is completely viable to black-box models.

**3. Unclear Terminology and Presentation:**
* The terms "constrained models" and "expansive models" appear in the abstract without definition, creating confusion about whether they refer to vocabulary strategies or model architectures.
* Section 7.2 examines different model scales within the InternVL family (1B, 2B, 8B, 14B) but concludes that "language model architecture significantly affects color vocabulary usage." This conflates model size with architectural differences.
* The paper assumes moderate familiarity with color perception research (Berlin & Kay, Munsell chips, focal colors) without adequate introduction for a general ML/AI audience.
* Typos: “thesaurus” in line 160.

**Questions:**

1. Can you provide statistical tests with p-values for claims of "significant" differences?

2. Why not include state-of-the-art commercial models (GPT-5, Claude, etc.) given that you mention "millions use them for everyday visual queries"?

---

### Author Response · Authors · 2025-11-21

We appreciate the reviewers’ helpful feedback. Given the scores, we’ve decided to withdraw the paper for now and use the feedback to make a stronger revision. Thanks again for your time and effort — it will really help us improve this work.

---

### Note · Authors · 2025-11-21

I have read and agree with the venue's withdrawal policy on behalf of myself and my co-authors.